# Improved Repeatability of Mouse Tibia Volume Segmentation in Murine Myelofibrosis Model Using Deep Learning

Aman Kushwaha *,†, Rami F. Mourad †, Kevin Heist , Humera Tariq, Heang-Ping Chan, Brian D. Ross, Thomas L. Chenevert, Dariya Malyarenko  and Lubomir M. Hadjiiski *

Department of Radiology, University of Michigan, Ann Arbor, MI 48109, USA
* Correspondence: amankush@umich.edu (A.K.); lhadjisk@umich.edu (L.M.H.)
† These authors contributed equally to this work.

**Abstract:** A murine model of myelofibrosis in tibia was used in a co-clinical trial to evaluate segmentation methods for application of image-based biomarkers to assess disease status. The dataset (32 mice with 157 3D MRI scans including 49 test–retest pairs scanned on consecutive days) was split into approximately 70% training, 10% validation, and 20% test subsets. Two expert annotators (EA1 and EA2) performed manual segmentations of the mouse tibia (EA1: all data; EA2: test and validation). Attention U-net (A-U-net) model performance was assessed for accuracy with respect to EA1 reference using the average Jaccard index (AJI), volume intersection ratio (AVI), volume error (AVE), and Hausdorff distance (AHD) for four training scenarios: full training, two half-splits, and a single-mouse subsets. The repeatability of computer versus expert segmentations for tibia volume of test–retest pairs was assessed by within-subject coefficient of variance (%wCV). A-U-net models trained on full and half-split training sets achieved similar average accuracy (with respect to EA1 annotations) for test set: AJI = 83–84%, AVI = 89–90%, AVE = 2–3%, and AHD = 0.5 mm–0.7 mm, exceeding EA2 accuracy: AJ = 81%, AVI = 83%, AVE = 14%, and AHD = 0.3 mm. The A-U-net model repeatability wCV [95% CI]: 3 [2, 5]% was notably better than that of expert annotators EA1: 5 [4, 9]% and EA2: 8 [6, 13]%. The developed deep learning model effectively automates murine bone marrow segmentation with accuracy comparable to human annotators and substantially improved repeatability.

**Keywords:** attention-U-net; myelofibrosis; MRI; test–retest pairs; volume wCV; mouse tibia segmentation





## 1. Introduction

Myelofibrosis (MF) is a chronic, ultimately fatal myeloproliferative neoplasm caused by genetic mutations in hematopoietic stem cells that cause systemic inflammation and progressive fibrosis, disrupting normal architecture and composition of the bone marrow [1,2]. Bone marrow biopsy remains the standard of care to assess the primary site of disease in MF [2,3]. These biopsies are painful and inherently suffer from sampling error, as the technique only analyzes millimeter amounts of tissue from one anatomic site (iliac crest) and do not assess heterogeneity of disease [4]. Development of quantitative imaging biomarkers (QIBs) of MF would facilitate comprehensive monitoring of disease progression and therapy response in patients in place of needle biopsies.

The goal of our ongoing co-clinical MF study [5,6] is to develop robust imaging protocols and analysis methods for quantitative imaging biomarkers of myelofibrosis. Investigation of the murine model of MF disease for preclinical testing of new therapeutic interventions [7] allows direct exploration of correlations between MRI and bone marrow pathology as full bone marrow can be extracted and studied under microscope [6]. To determine quantitative thresholds for significant changes in MRI biomarkers related to MF pathology, their measurement errors must be evaluated from repeatability studies [8,9] that include all the acquisition and analysis steps. Establishing QIB confidence intervals allows

detection of biological changes above the random measurement errors. Currently, "test–retest" experiments are standard means to estimate QIB repeatability in clinical oncology imaging studies, e.g., of lung, prostate, and breast cancer [10–13]. Following this practice, test–retest scans were collected for the preclinical arm of the MF study to assess bone marrow QIB repeatability [14].

The first essential step of the MF QIB analysis is segmenting the bone marrow within which QIB is measured. Manual segmentation by human operators is typically a tedious and time-consuming procedure and is prone to inter- and intra-observer variability [15,16]. Recently developed deep learning (DL) approaches show promise in both saving human effort and improving image segmentation accuracy and reproducibility [17,18]. One of the robust tools for image segmentation in the field of biomedical imaging is a U-Net based on deep convolutional neural network (DCNN) [19–21]. Convolution-based segmentation models take advantage of the spatial nature of images and learn well from relatively small training datasets common for medical imaging due to a shortage of available comprehensive expert annotations [18,21]. Thus, DCNNs with U-Net-based architectures currently are widely used for medical image segmentation [21,22]. In the current study, we evaluated the Attention U-Net (A-U-Net) consisting of a U-Net incorporated with attention gates [23] that can automatically learn to focus on important image features.

Key aspects where DL tools could help our ongoing co-clinical MF project are (1) to improve segmentation efficiency through automation of an otherwise tedious task for human experts, and (2) to improve precision of QIB by producing more repeatable segmentation results that serve as key input to QIB analysis. To explore the capability of DL tools for the above tasks, our current study focuses on assessing the A-U-Net model segmentation repeatability and relative accuracy compared to human expert segmentation of murine tibia, as well as the estimation of practical training size that can provide robust model performance.

## 2. Materials and Methods

The current study sought to determine whether the A-U-Net DCNN model segmentation of mice tibia would be more repeatable than manual segmentation without compromised accuracy. We performed test–retest repeatability and relative accuracy analysis on tibial bone volumes (scanned on two consecutive days) obtained by a deep learning (DL) segmentation model with respect to two human experts. This study design assumes that the tibia volume of a given mouse does not change significantly over the period of 1 day so that repeatability of the segmentation process can be assessed objectively without knowledge of the true tibia volume.

### 2.1. Experimental Design and Dataset

The studied dataset consisted of 3D Gradient echo (3D-GRE) MR images for 32 mice (26 diseased and 6 normal controls) with a total of 157 scans. The scans were acquired over a three-month period after induction of myelofibrosis by bone-marrow ablation for female mice with JK1 mutation. Forty-nine test–retest scan pairs were collected on consecutive days to assess repeatability [8,9]. The MRI acquisition parameters are detailed in Appendix A. Conversion from the acquired scanner native 3D magnitude-valued *2dseq*-image format to the meta-image header (MHD) ITK format was performed using custom routines in MATLAB (R2016b+, Mathworks, Natick MA, USA). The resulting MHD 3D volumes had a dimension of $128 \times 64 \times 256$ voxels with voxel size of $0.09 \times 0.075 \times 0.094$ mm$^3$ (voxel volume of $6.35 \times 10^{-4}$ mm$^3$) and gray-scale intensity bit-depth of 14; indexing from each dimension provided different cross-sectional views (Figure 1, inset).

The MHD images were used for DL model training and data analysis. The experimental design is shown in Figure 1. The available dataset was split into training (18 diseased mice, five controls: 107 scans), validation (two diseased mice, one control: 17 scans), and test (six diseased mice: 33 scans) subsets to approximately balance by scan number (70%/10%/20%). The validation and test subsets were not altered during evaluation. Due

to the small number of normal mice and our intended application for disease segmentation, their scans were included only in training (21 scans) and validation (six scans) sets. The full training set included 32 test–retest pairs, validation set contained four pairs, and test set had 13 pairs. To assess model sensitivity to the training set (TS) size, the full TS was split into three training subsets (Figure 1). Training split 1 (TS1) and training split 2 (TS2) each contained roughly half of the training data. Each split was balanced on the basis of test–retest pairs. The training single mouse (TSM) subset consisted of a single mouse randomly selected with a long time-series of (six biweekly scans), with four test–retest pairs.

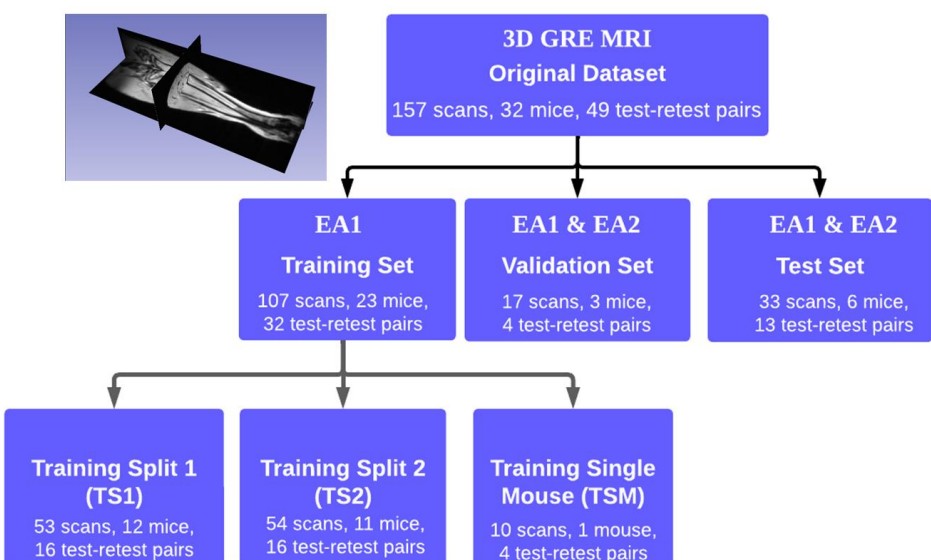

**Figure 1.** Experimental design diagram showing data content (including repeatability scan pairs) for different A-U-net "training" split scenarios (lower left branches). Data subsets that had manual segmentations by Expert Annotator 1 and 2 (EA1 and EA2) available are marked in the middle branches. The inset shows an example of 3D MRI image for mouse tibia with three orthogonal view projection planes.

Two expert annotators provided the tibia segmentation contours for the test and validation subsets. The masks contoured by Expert Annotator 1 (EA1) with 3 years of experience were used for training and as a reference for model accuracy assessment. Expert Annotator 2 (EA2) was instructed by EA1 (on a single-mouse example) to perform segmentations of the test and validation subsets. Scan and rescan annotations were performed independently. Prior to instruction, EA2 also dispatched the (unbiased) data subset selection for DL model training, validation, and testing. The reference tibia contours were drawn on the coronal view typically spanning 15–30 slices; thus, the coronal view was selected for segmentation by the DL model. The dimension of each grayscale MHD image in coronal view was 128 × 256, thus providing 64 2D images per mouse scan.

## 2.2. DL Model Architecture

We selected the A-U-net DL model for the tibia segmentation based on improved performance observed in our preliminary comparison study [24]. Our U-Net was implemented using Pytorch library (version 1.13.0) The base U-Net model architecture is detailed in [19,25]. Attention U-Net adds attention gates to the U-Net architecture [19,23] to increase the weights for automatically detected important image features, as illustrated in Figure 2. Each MRI slice was treated as an independent image, and the batch size (N) was randomly sampled from all available slices. The size of the input was (N,1,H,W) for a batch size of N, image height H, and width W. Attention U-Net consists of four encoder blocks and four decoder blocks (Figure 2A). The encoder network (contracting path) preserves the spatial

dimensions and doubles the number of filters after each stage whereas the decoder network (expanding path) doubles the spatial dimensions and halves the number of filters.

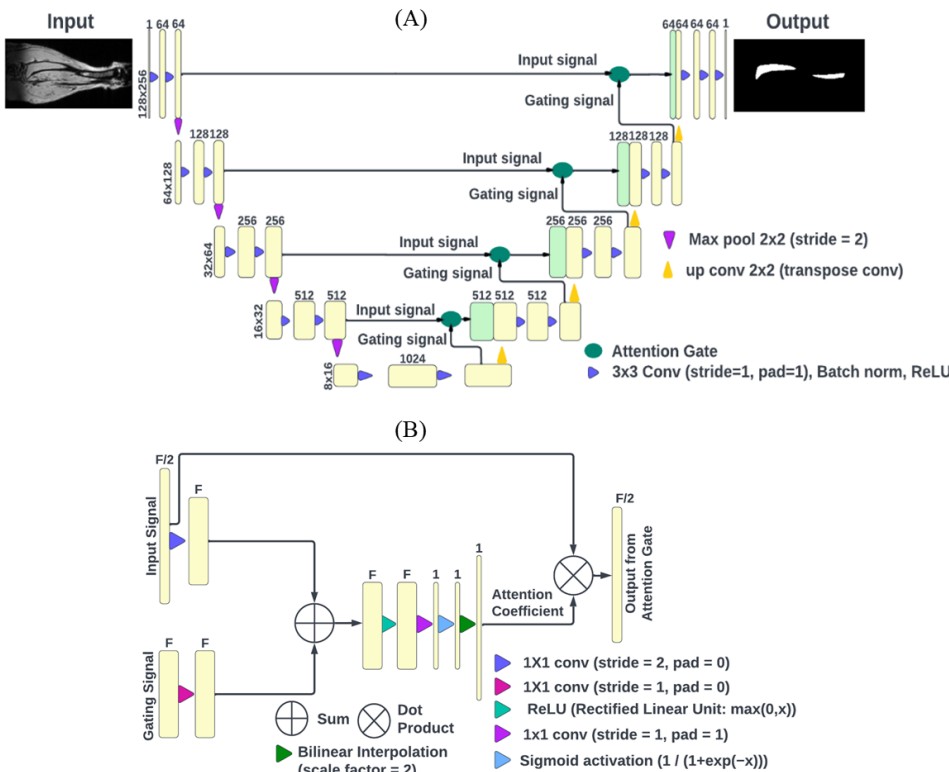

**Figure 2.** (**A**) Attention U-net architecture with (**B**) attention gate diagram: the black arrows indicate the workflow direction; yellow boxes depict the operation output (with the image dimensions (height × width) on the side and the number of applied filters shown on top), and green boxes correspond to output from the attention gate (**B**); colored triangles denote operations described in the legends. Output from the attention gate (green box) and up convolutions (yellow triangle) are concatenated (merged boxes). Attention coefficients are 2D arrays obtained from the input 2D MRI slices by the set of operations shown in (**B**).

The attention gate (Figure 2B) was described in [23]. Briefly, it takes the input signal (shape: (N,F/2,2H,2W)) and gating signal (shape: (N,F,H,W)) from two different stages, where the input signal and the gating signal have different sizes. The convolutions applied to the input signal (filter size = 1 × 1, stride 2, output size = F) and the gating signal (filter size = 1 × 1 and stride 1, output size = F) equalize their sizes, which are then added, and the rectified linear unit (ReLU) activation function is applied, followed by convolution (filter size = 1 × 1, stride =1, pad = 1, output size = 1) and sigmoid activation. Next, bilinear interpolation is applied to double the height and width to match that of the input signal, producing an attention coefficient array (N,1,2H,2W). The dot product of the attention coefficients with the input signal is the output from the attention gate that emphasizes the important features for image segmentation in the expanding path of the network (Figure 2A).

We experimented with training the Attention U-Net on the complete subset (Figure 1, TS), 50% splits (Figure 1, TS1 and TS2), and a single mouse subset (Figure 1, TSM) with respect to the EA1 segmentations as reference. The model was trained to minimize the binary cross-entropy loss. We fine-tuned the hyperparameters and selected the best model using the validation dataset for the different training scenarios. The model training typically took up to 120 epochs with 10,500 iterations for TS and batch size of 40 (6 h for TS, 3 h for TS1 or TS2, and 20 min for TSM using NVIDIA 1080Ti.) The best models were then

deployed to the held-out test set, and the automated segmentations were compared to the manual EA1 and EA2 segmentations.

### 2.3. Accuracy and Repeatability Evaluation

The DL model performance was assessed by segmentation accuracy with respect to EA1 reference and repeatability from test–retest experiments. Since true tibia volumes were unknown, only "relative accuracy" was assessed by this procedure. For an evaluation of the relative model accuracy, we used four commonly used performance metrics [19], namely, average Jaccard index (AJI), average volume intersection ratio (AVI), average volume error (AVE), and average Hausdorff distance (AHD) to compare the performance of the models and expert annotators (EA1 and EA2). The metrics are defined in Appendix B. The average of a given metric is the average over the entire set. Student's t-test with Bonferroni correction was applied to compare accuracy metrics (with respect to EA1 reference) of test-set mouse segmentations for different training scenarios (Figure 1, full training set, TS1, TS2, and TSM), and with EA2. The differences were considered significant for $p$-value < 0.009.

The relative accuracy of the model volume segmentations for the test subset (Figure 1) was also assessed by comparison to the EA1 reference via Bland–Altman (BA) analysis [9] and the agreement was quantified by Pearson correlation, R. BA analysis was also used to evaluate test–retest repeatability for volume segmentations by individual expert annotators and DL models trained on different data subsets. Bland–Altman plots provided a graphic view of agreement.

For the scan pairs acquired on consecutive days, the volume of the mouse tibia is not expected to change significantly; thus, the differences in the segmented volumes should ideally be zero. The repeatability and comparability of volumes estimated from the test–retest pairs were assessed using the within-subject coefficient of variation (wCV%) and BA analysis for limits of agreement (LOA) [9,26]. wCV is a relative (dimensionless) repeatability metric with confidence intervals (CIs) defined in Appendix B. The bias and LOA together determine the similarity between the test and retest volume pair. Small bias and narrow LOA indicate that the two measurements are essentially equivalent.

## 3. Results

### 3.1. DL Model Segmenation Accuracy with Respect to Reference

An example of the tibia segmentation from the validation set is presented in Figure 3. A darker tibia bone boundary is clearly visible. All the models accurately predicted the tibia bone contour with very minimal variations from the expert annotations even in regions of signal heterogeneity along the tibia length (Figure 3). The change in the tibia volumes over a period of 6 weeks (Figure 3E) was consistent with minor changes in the contours.

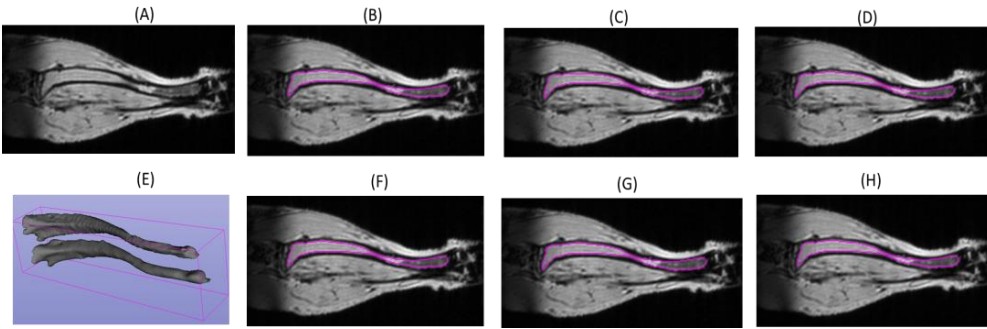

**Figure 3.** Examples of mouse tibia segmentation contours (magenta) for a single-slice MRI image in the test set (**A**) delineated by expert annotators ((**B**) EA1; (**F**): EA2) and by DL model with different training scenarios; (**C**) full training subset; (**D**) TS1; (**G**) TSM; (**H**) TS2. (**E**) The full tibia 3D volume segmentation by EA1 for two scans (6 weeks apart) of the same mouse (top = 6.5 mm$^3$; lower = 9.9 mm$^3$).

As expected, the DL model performance metrics (Table 1) reflect the highest relative accuracy (AJI: 86%–88%; AVI: 93%–97%) for the training sets. As the training set size is reduced, the standard deviations (SDs) get wider (e.g., from 5% to 10%) for the test subset segmentations.

**Table 1.** Summary of segmentation accuracy metrics (mean $\pm$ standard deviation (SD)) for both experts and DL model with different training split scenarios (full, split 1, split 2, and single mouse).

| | Dataset | AJI% | AVI % | AVE% | AHD (mm) |
|---|---|---|---|---|---|
| **Attention U-Net trained on full training set** | **Training** | $88.63 \pm 1.21$ | $94.64 \pm 1.64$ | $-1.44 \pm 3.07$ | $0.20 \pm 0.05$ |
| | **Validation** | $83.08 \pm 2.68$ | $91.06 \pm 3.79$ | $-0.67 \pm 6.28$ | $0.52 \pm 0.30$ |
| | **Test** | $83.45 \pm 5.11$ | $90.21 \pm 6.28$ | $1.71 \pm 7.94$ | $0.47 \pm 0.43$ |
| **Attention U-Net trained on split1** | **Training** | $86.17 \pm 2.22$ | $93.51 \pm 2.74$ | $-2.06 \pm 4.92$ | $0.39 \pm 0.68$ |
| | **Validation** | $81.47 \pm 4.58$ | $89.27 \pm 6.18$ | $1.22 \pm 8.65$ | $0.57 \pm 0.28$ |
| | **Test** | $82.47 \pm 6.79$ | $89.15 \pm 8.17$ | $2.83 \pm 9.89$ | $0.79 \pm 1.39$ |
| **Attention U-Net trained on split2** | **Training** | $88.34 \pm 1.15$ | $94.79 \pm 1.57$ | $-2.11 \pm 3.19$ | $0.23 \pm 0.06$ |
| | **Validation** | $82.78 \pm 2.65$ | $91.15 \pm 3.85$ | $-1.29 \pm 6.99$ | $0.78 \pm 0.77$ |
| | **Test** | $83.27 \pm 5.01$ | $89.94 \pm 6.39$ | $2.11 \pm 8.35$ | $0.49 \pm 0.42$ |
| **Attention U-Net trained on single mouse** | **Training** | $86.38 \pm 0.69$ | $97.32 \pm 1.09$ | $-9.99 \pm 2.97$ | $0.29 \pm 0.22$ |
| | **Validation** | $76.42 \pm 6.49$ | $87.70 \pm 8.45$ | $-2.41 \pm 11.63$ | $1.45 \pm 1.41$ |
| | **Test** | $77.99 \pm 7.43$ | $87.77 \pm 10.05$ | $-0.15 \pm 14.31$ | $1.10 \pm 1.07$ |
| **EA1 vs. EA2 (EA2 reference)** | **Training** | NA | | | |
| | **Validation** | $77.98 \pm 2.63$ | $97.39 \pm 1.39$ | $-22.44 \pm 6.46$ | $0.36 \pm 0.15$ |
| | **Test** | $80.70 \pm 2.91$ | $96.70 \pm 2.33$ | $-16.70 \pm 7.66$ | $0.29 \pm 0.11$ |
| **EA2 vs. EA1 (EA1 reference)** | **Training** | NA | | | |
| | **Validation** | $77.98 \pm 2.64$ | $79.70 \pm 3.36$ | $18.11 \pm 4.39$ | $0.36 \pm 0.15$ |
| | **Test** | $80.70 \pm 2.91$ | $83.10 \pm 3.92$ | $13.94 \pm 5.76$ | $0.29 \pm 0.11$ |

AJI: average Jaccard index, AVI: average volume intersection ratio, AVE: average volume error, AHD: average Hausdorff distance.

However, the average performance metrics are sufficiently close (within SD) for the TS1, TS2, and full training subsets, and similar accuracy is observed for the test and validation subsets when the models were trained on half of the training data. Compared to training with the full set, the accuracies and performance metrics for the test subset were only slightly lower (<1.5%) when the models were trained on TS1 or TS2, which was not significantly different after Bonferroni correction ($p > 0.015$). For the test subset, the relative accuracy of DL model trained on a single mouse was significantly lower compared to the other training scenarios. The AHDs were below 1 mm for all training scenarios, increasing with decreasing training set size.

The lower value of AJI% for EA2 vs. EA1 reflects the interobserver variability in the way EA1 and EA2 generate the masks. There is large over-segmentation by EA2 relative to EA1 as observed from the AVEs of 14–18%. The interobserver variability in mouse tibia segmentation by the human experts is much larger than that of the A-U-Net models relative to EA1 (<3%). For the test set, the relative accuracy of EA2 segmentations with respect to the EA1 reference (AJI = 81%, AVI = 83%, AVE = 14%) was notably below that achieved by A-U-net models trained on full or TS1 and TS2 subsets (AJI = 82.5–83.5%, AVI = 89–90%, AVE = 2–3%), with significant improvement for AVI (6%) and AVE (−11%).

In the test example shown in Figure 4, the DL model segmentations derived after training with full and half datasets (Figure 1, TS1 and TS2) apparently produced slightly more accurate tibia segmentation at the distal end compared to somewhat under-segmented EA1 "reference" (Figure 4B, yellow arrow). Since expert segmentation may have variabilities, the agreement or disagreement with respect to the reference annotations may not reflect the true model accuracy (Table 1). The predictions made by all the models (Attention U-Net

trained on multiple mice) are fairly robust except for the one trained on a single mouse (Figure 4G). The predicted tibia mask by the model trained on the TSM subset misses a large portion of bone contour near the knee (indicated by the yellow arrow in Figure 4G). Thus, the model learned from a small training set shows declining performance and is more susceptible to errors in reference annotations of the training samples. The small data size fails to include all the diversity and features that would be encountered while making the test predictions.

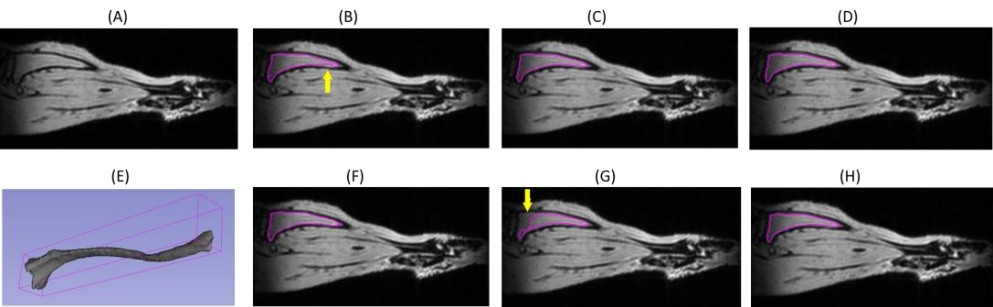

**Figure 4.** Examples of mouse tibia segmentation contours (magenta) for a single-slice MRI image in the test set (**A**) delineated by expert annotators ((**B**) EA1; (**F**) EA2) and by DL model with different training scenarios (**C**) full training; (**D**) TS1; (**G**) TSM; (**H**) TS2. (**E**) The full tibia 3D volume (6.5 mm$^3$) segmentation by EA1. The yellow arrows indicate apparent tibia under-segmentations in (**B**) and (**G**).

Figure 5 further summarizes the relative accuracy of volume segmentations in the test subset with respect to the EA1 reference for different A-U-Net training scenarios. EA2 segmentations show systematic positive bias ~1 mm$^3$ (consistent with tendency to over-segment) while the A-U-Net segmentations are essentially free of bias, which indicates that the models learn to segment in a similar way as EA1 as they were all trained with the EA1 annotations. Except for single-mouse training (TSM) (dispersion ~2 mm$^3$, R ~0.74, all training scenarios are in good agreement with EA1 reference (dispersion <1 mm$^3$ and model-generated volumes highly correlated to the reference: R ~0.92 to 0.95). The DL models are also highly correlated among themselves (R ~0.98 to 0.99). The notable decline in relative model precision for TSM confirms the insufficient training set size (Figure 1, TSM).

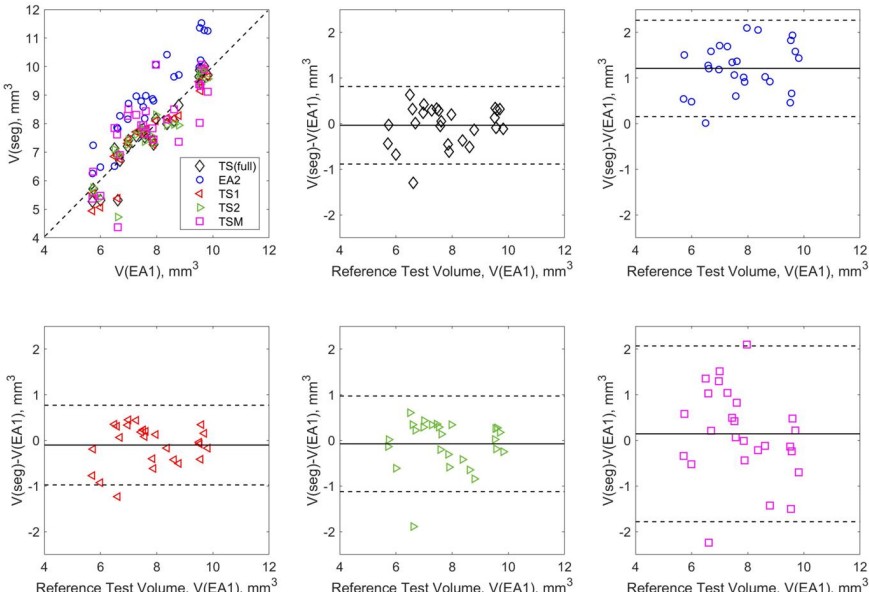

**Figure 5.** Tibia volume (including test–retest) accuracy for the test subset with respect to EA1 reference for different training scenarios (color-coded in the legend). Solid horizontal lines mark the bias (average volume difference), and dashed lines correspond to 95% limits of agreement.

### 3.2. DL Model Segmenation Precision

Table 2 further summarizes the observed repeatability trends in volume estimations for the test–retest pairs in all data subsets and volumes. For single-mouse training and all validation results that included only four test–retest pairs (Figure 1), the CI range is quite broad (from 6% to 35%), likely resulting from insufficient training to handle the large variations in image properties of unknown validation and test cases and, thus, causing inconsistent segmentation of the tibia volumes. As expected, for the 16–32 test–retest pairs in the training subsets, the model-derived tibia volume repeatability is close to that of the EA1 reference with an average wCV = 7.7% and CI = 5.5–12.3%, as the A-U-Net models were specifically trained to follow the EA1 segmentations.

**Table 2.** The repeatability of volume estimations in terms of %wCV and confidence intervals [CIs] in all data subsets for A-U-net model and EA1, EA2 expert annotations and different DL model training scenarios (Figure 1: full, TS1, TS2, TSM).

| Model | | Training | Validation | Test |
|---|---|---|---|---|
| | | wCV [CI] % | | |
| **A-U-net** | **Full** | 7.7 [6.2, 10.3] | 2.0 [1.1, 7.4] | 2.6 [1.9, 4.3] |
| | **TS1** | 7.6 [5.5, 12.0] | 5.3 [3.0, 20.0] | 3.1 [2.2, 5.1] |
| | **TS2** | 8.0 [6.0, 12.3] | 1.83 [1.0, 6.8] | 3.2 [2.3, 5.3] |
| | **TSM** | 3.3 [2, 12] | 11.1 [6.3, 41.6] | 7.0 [5.0, 11.6] |
| **EA1** | **Full** | 7.7 [6.2, 10.3] | 4.4 [2.5, 16.3] | 5.3 [3.8, 8.7] |
| | **TS1** | 7.3 [5.4, 11.5] | | |
| | **TS2** | 8.0 [6.0, 12.3] | | |
| | **TSM** | 5.8 [4.6, 7.7] | | |
| **EA2** | **Full** | NA | 2.5 [1.4, 9.1] | 8.0 [5.7, 13.2] |
| | **TS1** | | | |
| | **TS2** | | | |
| | **TSM** | | | |

Similar precision performance is observed for EA2 on the test set with wCV = 8%, while EA1 with wCV = 5.3% is notably better, but there is substantial CI overlap (3.8–8.7% versus 5.7–13.2%), reflecting inter- and intra-observer variabilities for manually segmented tibia volumes. In contrast, for DL models trained on the full set, TS1 or TS2, the average repeatability (wCV ~3%) with tight CI = 1.9–5.3% is about twofold better than that for single-mouse training or for EA1 and EA2 segmentations of the test–retest pairs in the test set.

The repeatability of tibia volume measurements for all DL models and expert annotators is compared for the test–retest pairs in the test subset in Figure 6. The models show negligible (positive) volume bias between the scan and rescan segmentations, while EA2 shows notable positive bias (0.4 mm$^3$), consistent with the tendency of EA2 to slightly over-segment for the repeated scan (similar to over-segmentation relative to EA1 as observed in Figure 5). The LOAs for tibia volumes generated by DL models trained on all or half the data are small with the overall precision within 0.5 mm$^3$. The precision of EA1 is about half of DL model with the spread up to 1 mm$^3$. For the EA2 and DL model trained on a single mouse, the precision is threefold lower, with an overall spread of about 1.5 mm$^3$.

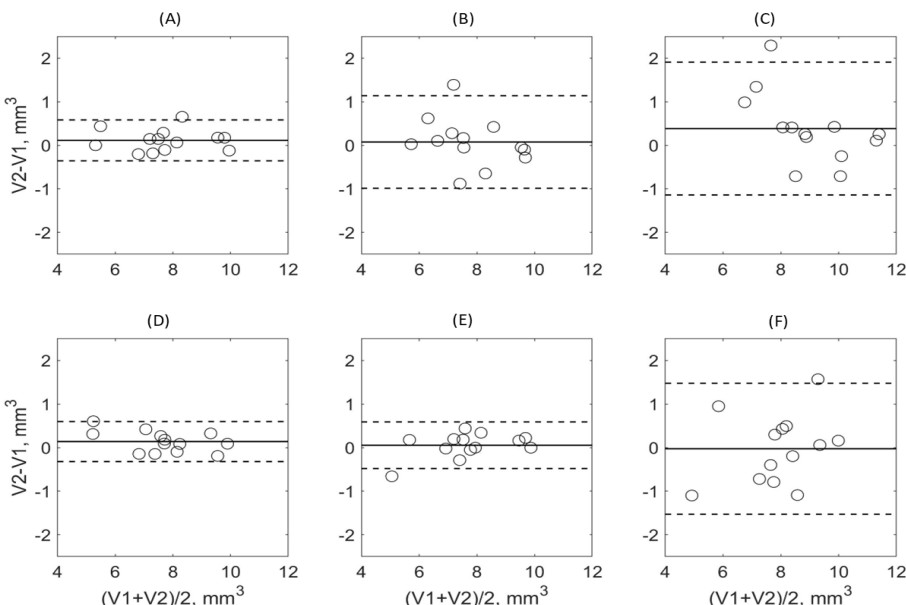

**Figure 6.** Bland–Altman (BA) analysis of the scan–rescan (test–retest) tibia volumes in the test dataset for different A-U-net training scenarios ((**A**) full training set; (**D**) half-split 1; (**E**) half-split 2; (**F**) single-mouse) and human expert annotations ((**B**) EA1; (**C**) EA2) segmentations. Solid horizontal lines mark the observed average bias of the volume difference, while dashed lines mark corresponding 95% limits of agreement.

## 4. Discussion

The main finding of our study is that DL-based segmentation shows strong promise in improving repeatability of tibia volume measurements while the accuracy is comparable to manual annotations. For the test subset, the accuracy of DL segmentation trained on more than 10 animals (>52 scans) notably exceeded that of the second expert annotator (by 6% for volume intersect ratio and by −11% for volume error), with repeatability errors reduced to 3% from 5%–8% (observed for manual segmentations). Thus, the developed model has strong potential to enhance precision of the corresponding volume-based bone marrow QIBs of myelofibrosis [6] with substantial saving of human effort. The utilized test–retest performance metrics is generally more objective than relative accuracy assessment and would be useful for routine evaluation of DL model-based segmentation precision.

The variability in volumes from manual segmentation is a known major factor that limits precision in quantitative imaging [12,13,27]. The manual image segmentation is a tedious time-consuming task with repeatability dependent on the level of experience, attention span and fatigue of an annotator. Variabilities among the experts are also known to contribute to reproducibility errors [15,16]. Since human annotations are generally not free from subjective judgments, only relative segmentation accuracy can be assessed for the DL models with respect to the expert-provided reference. In fact, we noticed that the DL model trained on multiple mice can provide more accurate segmentations compared to DL model trained on a single mouse, while lower performance measures could just indicate discrepancy with the expert reference. In contrast, the segmentation repeatability is a more objective measure of model performance since the tibia volumes should not change between test and retest scans.

For clinical QIB precision assessments, QIBA recommends a test–retest sample size of >35 [8] to evaluate wCV with nominal CIs. Our preclinical validation sample size was insufficient for confident wCV measurement, but the samples in the test subset provided moderate CIs that allowed confident detection of twofold improved repeatability of DL model-based mouse tibia segmentations (3%) compared to expert annotations (5.3% to 8%) with relatively high accuracy (>82%) when the model was trained on the dataset of the size exceeding test dataset size. Apparently, training on 53–54 scans for 11–12 animals enabled

relative accuracy and repeatability for test set segmentation equivalent (within confidence intervals) to training on 107 scans for 23 animals. Small image datasets with limited expert annotations are typical for biomedical image segmentation applications [28,29]. The training sample size required for robust model training is expected to depend on the complexity of the task and the DL architectures used. Therefore, estimating reasonable size for DL model training for a given task is of practical value to save human effort and improve segmentation consistency [30,31].

To date DL segmentation of murine MRI data has predominantly focused on application to soft-tissue regions and lesions [32,33]. In contrast to prior DL segmentation studies of murine bone primarily performed for microcomputed tomography (μCT) images [30,34,35], our current study provides the first example of DL bone segmentation in high-resolution MRI data. We studied the use of 2D images and the dependence on training sample sizes for training A-U-Net models. The relative accuracy of 80–90% and AHD <1 mm achieved for bone segmentation are similar to those reported for the μCT studies [30,34,35]. Although prior DL segmentation work compared relative model performance for multiple annotators [34] and different CT scanners [35], none used scan–rescan analysis for objective precision assessment as reported in our study.

Our study had limitations. First, the test and training subsets (of relatively small sizes) came from the same dataset acquired on a single scanner with fixed scan protocol for a single mouse mutation model; second, the reference annotations were provided by a single expert. These would likely limit direct generalization of the model for different acquisition protocols and should be tested independently for different murine models. However, the implementation of the automated segmentation provided valuable insights into practical DL model-based segmentation workflow, which could be utilized for other quantitative MRI acquisition protocols and image datasets.

The mouse tibia anatomy is largely consistent among the animals, allowing the use of highly uniform imaging protocols. Thus, mouse tibia segmentation may represent a relatively straightforward task for DCNN model, providing robust example for development of performance evaluation workflow. Apparently, the Attention U-Net trained on more than 50 scans successfully handled the challenges of both heterogeneous intensities and small murine bone sizes. The developed DL model for murine tibia bone marrow segmentation can be directly applied to future mouse scans acquired with similar MRI protocols. In future work, iterative semi-supervised DL training could be implemented for murine bone segmentation. In such an approach, manual segmentation is performed for a small subset of mouse scans for model training, followed by automatic segmentation and quality check for a small batch of new cases and adding manually corrected (challenging) cases to retrain the model on the enlarged annotated set. The process can be iterated for a number of times until the performance of the trained model is acceptable. This approach may substantially reduce the effort of manual segmentation.

Our upcoming myelofibrosis studies will focus on generalizing the DL model workflow for human bone marrow segmentation. The human bone sizes are much larger, and expert annotations are even more time-consuming. Additionally, there is intrinsically higher variability of orientations, anatomic details, and scan parameters compared to mice, which might potentially require a larger training dataset and or transfer learning. Overall, improved processing throughput is vital for development, validation, and implementation of MRI-based quantitative biomarkers for advancing experimental therapeutics in myeloproliferative neoplasm mouse models with the overarching goal of improving patient outcomes.

## 5. Conclusions

DL-based segmentation shows strong promise of improved repeatability of murine tibia volume measurements from MRI scans with accuracy comparable to manual annotations and μCT. Attention U-Net was found to be an efficient alternative to human expert

segmentations of tibia, providing high accuracy and precision for quantitative imaging of myelofibrosis.

**Author Contributions:** Conceptualization, T.L.C., D.M. and L.M.H.; methodology, L.M.H., H.-P.C. and D.M.; software, A.K. and L.M.H.; validation, A.K., R.F.M. and H.T.; formal analysis, A.K. and H.T.; investigation, K.H.; resources, B.D.R. and K.H.; data curation, K.H. and R.F.M.; writing—original draft preparation, A.K., R.F.M., H.T. and D.M.; writing—review and editing, T.L.C., B.D.R., K.H., H.-P.C. and L.M.H.; visualization, A.K., R.F.M. and D.M.; supervision, T.L.C., D.M. and L.M.H.; project administration, B.D.R. and T.L.C.; funding acquisition, B.D.R. and T.L.C. All authors have read and agreed to the published version of the manuscript.

**Funding:** This research was funded by National Institute of Health grant numbers R01CA238023, U24CA237683, R01CA190299, U01-CA232931, and R35CA197701.

**Institutional Review Board Statement:** The animal study protocol was approved by the Institutional Ethics Committee of University of Michigan (protocol code 00006795 approval date:01/21/2016).

**Informed Consent Statement:** Not applicable.

**Data Availability Statement:** The mouse MHD image and annotation collection and tibia volume tables are available upon request.

**Conflicts of Interest:** The authors declare no conflict of interest. The funders had no role in the design of the study; in the collection, analyses, or interpretation of data; in the writing of the manuscript; or in the decision to publish the results.

## Appendix A. MRI Acquisition Parameters

All mice were scanned using a fixed 3D FLASH coronal-plane imaging protocol on 7 T 30 cm bore Agilent system magnet with Small Receive CryoProbeTM4-Element Array RF Coil Kit, cryogenically cooled to 20–30 K. The anesthetized mouse leg was held in place between a 3D-printed, leg-shaped mold on posterior side and the CryoProbeTM on anterior side for a 15 min scan. The imaging protocol parameters were defined in Bruker BioSpec® MRI Console Paravision 7.0.0 software and included single-echo; TR/TE = 111 ms/2.99 ms; flash-spoiling; flip angle = 9°; 1 NSA; imaging matrix size: $256 \times 128 \times 64$; voxel size: $0.09 \times 0.075 \times 0.094$ mm$^3$; field of view (FOV): $23 \times 9.6 \times 6$ mm$^3$.

## Appendix B. Performance metrics definitions

Jaccard index: It is defined as the ratio of the intersection of the segmentation volumes divided by the union of the volumes of the two 3D masks. The Jaccard index for a single scan is calculated as

$$\text{Jaccard Index} = \frac{V_h \cap V_c}{V_h \cup V_c},$$

where $V_h$ and $V_c$ are the volumes of the segmentation mask drawn by human as reference standard and predicted by computer, respectively, for the same scan.

Volume intersection ratio: The volume intersection ratio for the segmentations of a single scan is calculated as

$$\text{Volume Intersection Ratio} = \frac{V_h \cap V_c}{V_h},$$

where $V_h$ and $V_c$ are defined as above.

Volume error (AVE): The volume error for the segmentations of a single scan is calculated as

$$\text{volume error} = \frac{V_h - V_c}{V_h},$$

where $V_h$ and $V_c$ are defined as above. A positive value of volume error indicates under-segmentation, and a negative value indicates over-segmentation by the computer.

Hausdorff distance (AHD):

$$\text{HD} = max\{ \ max_{x \in r1}\{ \ min_{y \in r2}\{ \ d(x,y) \ \} \}, \ max_{y \in r2}\{ \ min_{x \in r1}\{ \ d(x,y) \ \} \} \ \}.$$

The Hausdorff distance measures the maximum distances between the closest points of two segmentation contours, where r1 and r2 are set of points in 3D of the two contours respectively, and $d(x,y)$ is the Euclidean distance.

Within-subject coefficient of variance (wCV):

The sum and difference between the paired observations are conveniently used to calculate pairwise mean $M_i$ and pairwise variance $V_i$ for each $i^{th}$ test–retest pair as follows:

$$M_i = \frac{(V_1 + V_2)}{2},$$

$$V_i = \frac{(V_2 - V_1)^2}{2},$$

where $V_1\left(mm^3\right)$ and $V_2\left(mm^3\right)$ denote the volumes from the same tibia in scan 1 and scan 2, respectively, which are, thus, measured twice on different days yielding a single paired observation. The square of the within-subject coefficient of variation $\left(wCV^2\right)$ is first obtained by taking the mean of the ratio of variance $V_i$ to the square of the mean $M_i^2$ over all N pairs and then is applied to calculate the within-subject coefficient of variation (wCV%) as a percentage as follows:

$$wCV^2 = \frac{1}{N} \sum_i^N \frac{V_i}{M_i^2},$$

$$wCV\% = 100 \times wCV^2.$$

The corresponding 95% confidence intervals ($\alpha = 0.05$) to $wCV$ are given by the multiplicative lower-bound (LB) and upper-bound (UB) factors given by ChiSqr function for N-1 degrees of freedom as

$$95\% \text{ CI of wCV} = \ wCV\% \cdot \left[ \frac{1}{\sqrt{(N-1)\cdot \chi^2_{0.975}}}; \ \frac{1}{\sqrt{(N-1)\cdot \chi^2_{0.025}}} \right].$$

Bland–Altman analysis (BA):

The BA agreement was computed for the pairwise volume differences $(V_2 - V_1)_i$ against their means $M_i$. The mean of all N paired differences on tibia volume data is assessed to measure the bias between test and retest volumes. Ideally, the bias should be close to zero thereby supporting the assumption that the volume will remain constant between test and retest measurements.

$$Bias = \frac{1}{N} \sum_i^N (V_2 - V_1)_i.$$

The corresponding 95% limits of agreement (LOA) along the difference axis $(V_2 - V_1)$ in Bland–Altman plots provide a graphical indication of measurement precision. To find the precision, the standard deviation SD of the differences between test and retest volumes is obtained, and then the 95% LOA of $(V_2 - V_1)$ is calculated as follows:

$$Precision \ (LOA) = Bias \ \pm 1.96 * SD \ (V_2 - V_1).$$

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
