# Peer review of "Improved Repeatability of Mouse Tibia Volume Segmentation in Murine Myelofibrosis Model Using Deep Learning"

_tomography, doi:10.3390/tomography9020048_

Round 1

Reviewer 1 Report

The manuscript submitted by Dr., Aman Kushwaha has set up a deep learning model of MF in murine tibia to evaluate segmentation methods which can be used to assess disease status via image-based biomarkers. This is an interesting topic. The authors have used 32 mice with 157 MRI scans and 49 test pairs to set up the training, validation, and test subsets. The setting ratio of training and test is fine to me. However, the whole dataset is not big enough. Therefore, the accuracy of volume segmentations in the test subset reduced when the training set size decreased. In comparison with 3 models used in this manuscript, the authors found that DL-based segmentation showed a strong promise of improved repeatability of tibia volume measurements while the accuracy is comparable to manual annotations. 

I doubted whether the aim for this study is to find the model mimic the manual annotation or one better method which is more efficient than manual annotation. 

No model is perfect. However, I would like to see a bigger datasets with  a more efficient model which can save the technique's time and improve the accuracy. 

Author Response

We thank both Referees for their constructive comments and made clarifying changes (tracked) in the revised manuscript (Tomography-2186577) to address their comments (in italics), as highlighted in bold below.

Reviewer 1:

This is an interesting topic. The authors have used 32 mice with 157 MRI scans and 49 test pairs to set up the training, validation, and test subsets. The setting ratio of training and test is fine to me. However, the whole dataset is not big enough. Therefore, the accuracy of volume segmentations in the test subset reduced when the training set size decreased.

R1.C1: We agree with Reviewer that small training data set is a limitation of the study. We added explanation in Introduction and emphasize in Discussion that the lack of annotated dataset for DL algorithm training is a usual limitation of the current imaging studies. In fact, such data sets are particularly scarce for murine MRI imaging, and one of the goals of our study was to determine a reasonable training data size for sufficient accuracy of automated murine tibia segmentation. We were pleased to find that there was no statistically significant difference (<1.5%) for the accuracy metrics of the models trained on half-sized training set (11-12 animals and 53-54 scans) versus full training set (23 mice, 107 scans), and achieved accuracy exceeded that for manual segmentation by the second expert annotator. This point is now clarified both in the revised Abstract, Results and in Discussion.

I doubted whether the aim for this study is to find the model mimic the manual annotation or one better method which is more efficient than manual annotation

R1.C2: As is clarified in the Introduction and Discussion, the goal of this study was to build a practical workflow for DL model training to achieve sufficient accuracy and repeatability for murine tibia segmentation to replace human annotators in future murine tibia studies. We were pleased to find that the achieved accuracy for our first MRI study of murine tibia was comparable to prior DL bone segmentation available for micro-CT imaging modality, while repeatability was improved better than two-fold compared to manual segmentation.

No model is perfect. However, I would like to see a bigger datasets with a more efficient model which can save the technique's time and improve the accuracy

R1.C3: Since our study is the first MRI investigation of tibia segmentation, no larger data set with tibia annotations is currently available for training the model. As we explained in Discussion, to circumvent the sparsity of annotated data for the murine tibia imaging field, we plan to enact a semi-supervised retraining for the model with newly acquired data as project progresses.

Reviewer 2 Report

The paper describes an approach to facilitate studies of myelofibrosis in a murine model. In MR studies of bone marrow human expert evaluation is replaced by deep learning evaluation. “Level of expertise, attention span and fatigue” of human annotators shall bypassed this way. Quantitative imaging biomarkers would easily and reliably generated by a deep learning evaluation.

Two expert annotators delivered masks for training and model accuracy and the basis of repeatability assessment. Different training sets were generated and test-retest pairs are a central approach of the study. Estimation of a practical training size is a second goal.

In summary, the study hints at improved repeatably of deep learning when compared to humans and comparable accuracy. A reduction of size of training sets lead to a wider standard deviation of the quantitative imaging biomarker.

The study is well designed and the paper as a whole is easy to follow (I am not an expert on the informatics and cannot comment on this part). The usual limitations are laid open in line 339 to 346.

Two things could probably be improved: The abstract may be perceived as somewhat prohibitive by readers. While certainly correct, lines 19 to 26 are difficult to makes sense of at first reading; it might be preferable to just give a  summary. On the other hand, “streamlines bone segmentation” in line 26 could be rewritten using a more scientific language. – If I understood it right, the size estimation of training sets rest on single animal,  training split 1 and training split 2. The actual effects of the second and third set are difficult to extract from table 1. This would deserve a paragraph in the methods section.

In summary this is a diligently executed and  presented study of great practical value to a variety of fields.

Author Response

We thank both Referees for their constructive comments and made clarifying changes (tracked) in the revised manuscript (Tomography-2186577) to address their comments (in italics), as highlighted in bold below.

Reviewer 2:

The study is well designed and the paper as a whole is easy to follow (I am not an expert on the informatics and cannot comment on this part). The usual limitations are laid open in line 339 to 346. Two things could probably be improved: The abstract may be perceived as somewhat prohibitive by readers. While certainly correct, lines 19 to 26 are difficult to makes sense of at first reading; it might be preferable to just give a  summary. On the other hand, “streamlines bone segmentation” in line 26 could be rewritten using a more scientific language.

R2.C1: We appreciate Reviewers comments on the study design and helpful suggestions to clarify the presentation. As suggested, the abstract was modified to summaries the main findings and clarify the conclusions.

If I understood it right, the size estimation of training sets rest on single animal,  training split 1 and training split 2. The actual effects of the second and third set are difficult to extract from table 1. This would deserve a paragraph in the methods section.

R2.C2: As suggested, we added explanation of the performance assessment criteria for the investigated training scenarios (based on the accuracy and repeatability of “test” subset) in Methods and clarified Results for Table 1.